# Potential cross-species correlations in social hierarchy and memory between mice and young children

Yu-Ju Chou [1✉], Yu-Kai Ma[2], Yi-Han Lu[2], Jung-Tai King[3], Wen-Sheng Tasi[2,4], Shi-Bing Yang [4✉] & Tsung-Han Kuo [2,5✉]

Social hierarchy is associated with various phenotypes. Although memory is known to be important for hierarchy formation, the difference in memory abilities between dominant and subordinate individuals remains unclear. In this study, we examined memory performance in mice with different social ranks and found better memory abilities in dominant mice, along with greater long-term potentiation and higher memory-related gene expression in the hippocampus. Daily injection of memory-improving drugs could also enhance dominance. To validate this correlation across species, through inventory, behavioral and event-related potential studies, we identified better memory abilities in preschool children with higher social dominance. Better memory potentially helped children process dominance facial cues and learn social strategies to acquire higher positions. Our study shows a remarkable similarity between humans and mice in the association between memory and social hierarchy and provides valuable insight into social interactions in young animals, with potential implications for preschool education.

[1] Department of Early Childhood Education, National Tsing Hua University, Hsinchu 300, Taiwan, Republic of China. [2] Institute of Systems Neuroscience, National Tsing Hua University, Hsinchu 300, Taiwan, Republic of China. [3] Institute of Neurosciences, National Yang Ming Chiao Tung University, Taipei 112, Taiwan, Republic of China. [4] Institute of Biomedical Sciences, Academia Sinica, Taipei 115, Taiwan, Republic of China. [5] Department of Life Science, National Tsing Hua University, Hsinchu 300, Taiwan, Republic of China. ✉email: chouyuju@mx.nthu.edu.tw; sbyang@ibms.sinica.edu.tw; thkuo@life.nthu.edu.tw

The dominance hierarchy is a common social structure in several animal species. Under a stable hierarchical relationship, dominant animals have priority in the choice of resources, such as territory, food, and mating partners, whereas subordinate individuals receive fewer resources but are subjected to less conflict and peril[1]. Hierarchical ranking, therefore, greatly influences different behaviors, especially social behaviors, such as mating and aggression[2], as well as physiological components, such as stress hormones, cardiovascular function, and the immune system[3]. The causality or cross-talk between social rankings and different phenotypes is extremely complicated and remains to be further elucidated.

The conceptual framework of social hierarchy has been applied to human interactions and has revealed recognizable dominance hierarchies, even in preschool-aged children[4–7]. Observational studies have indicated that children with more aggressive behaviors are usually recognized as higher ranking in their group and are held in higher regard by their peers compared with subordinate children[8–11]. In addition to the tendency to be coercive, young children's adoption of and change in social strategies to reach their dominance goals are also the focus of child-development research[9,12,13]. Those who can gradually change from using coercive strategies to using prosocial or dual strategies to control resources are more likely to achieve higher status for a long time[9,12,13]. Cognitive-oriented studies in recent years have further indicated that human children first become aware of social-dominance cues in early infancy[14]. Young children aged 3–4-years old are able to use multiple physical and social cues to detect the social-dominance status of others (e.g., facial expressions, body size or body postures, and the interaction of characters in experimental films)[15–17]. At approximately 5 or 6 years of age, some children can even determine the relationships of dominance based only on static photographs[15]. Surprisingly, despite children's amazing learning ability concerning social dominance, research on factors related to social dominance has mostly focused on environmental factors, such as interaction experience, parenting style, moral education, and cultural background[9,12,18–20]. The neural mechanisms underlying how children recognize social-dominance cues and the underlying cognitive ability that supports social-strategy learning have rarely been studied.

Social hierarchies, from the neural mechanisms of dominant behaviors to the impacts of social status on a variety of phenotypes, have also been studied in laboratory-model organisms. For rodent species, including mice and rats, aggression was used to be the major assay to evaluate social status and explore the relationships of social hierarchy with innate behaviors, anxiety/depression-like behaviors, cognitive ability, and other physiological phenotypes[21–25]. However, because aggressive behaviors are primarily performed by adult males, the social hierarchies of females and young animals have rarely been explored[26–28]. However, the tube test, in which one mouse forces its opponent backward out of a tube, is relatively easy for mice to complete and has been used largely to study social dominance[29–31]. Since the assay only requires mice moving forward and backward inside the tube, it presents as a platform to investigate the relationship between social hierarchy and other traits not only in adults but also in young animals[28,32].

Mice have been widely used as a standard model organism for different human-biology fields, including psychological and neurological diseases[33–36]. However, parallel studies between humans and mice, especially in behavioral research, have been rare in the past. In recent years, neuroscientists have started to study these two species together and found surprising similarities, not only in behaviors but also in the underlying mechanisms[37]. Mouse studies could offer various techniques (including intrusive designs and manipulative approaches) to investigate the mechanistic questions that could not be conducted in human subjects; the verification and extended exploration in human studies further emphasized the biological significance of the discovery. The findings from these two species could, therefore, complement each other and provide important information from different perspectives. By taking advantage of this comparative and complementary approach to investigate social-dominance behaviors, our recent study demonstrated comparable social ranks between young human children and weanling mice and further revealed important intrinsic factors involved in the early formation of social hierarchy[32]. This study, however, did not identify the potential role of learning and memory in hierarchical formation in mice and children.

While the influence of social status on different behavioral or physiological characteristics has been investigated largely, the relationship between memory and social hierarchy has been much less explored. Studies in rats and anolis have suggested that memory could play a role in enhancing the stability of the hierarchical relationship[38,39]. Whether there is a difference between dominant and subordinate individuals in memory abilities, however, remains controversial and is an open question. From an evolutionary perspective, the ability to remember one's own social status or recognize others should be important, especially to aid subordinate animals in avoiding conflict[39–42]. On the other hand, better memory could potentially help dominant animals memorize surrounding environments or acquire new skills to control resources. Unfortunately, previous studies in mice were unable to provide a conclusive answer[25,43,44]. In human children, to our knowledge, this question has never been asked. To address this question, in this report, we examined memory performance in mice and children and found a positive correlation between social hierarchy and memory abilities. Mechanistically, dominant mice showed greater long-term potentiation (LTP) and higher expression of memory-related genes in the hippocampus than the subordinates. Improving mouse memory with sodium butyrate (SB) or rolipram was also found to enhance dominant status. Functionally, better memory in human children contributes to the learning and adoption of social strategies, as well as the neurocognitive processing of social-dominance cues for the acquisition of social dominance status. The findings of this study, therefore, provide important information to the research field of social interaction in both animals and human children, and have beneficial implications for early childhood education.

## Results

**High-rank mice exhibited better memory ability.** Consistent with our previous study[32], a social hierarchy can be established in 4 weanling mice in a cage by the tube test (Supplementary Fig. 1). To assess the correlation between social ranks and memory ability, we first conducted the novel object-recognition (NOR) test with a 1-hour interval between the training and test sessions to evaluate short-term recognition memory in weanling mice[45]. Surprisingly, only 1st- and 2nd-rank mice spent significantly more time exploring a novel object, but not 3rd- and 4th-rank mice (Fig. 1a), suggesting that, in contrast to dominant mice, subordinate mice had more difficulties recognizing a familiar object in the test. Analyses of the discrimination index also showed a higher trend of the index in 1st-rank mice than 4th-rank mice ($p = 0.077$) (Fig. 1b), and the index was negatively correlated with the four levels of social rank (Fig. 1c). The higher discrimination index in dominant mice was unlikely due to higher curiosity, since there was no correlation between social ranks and exploratory behavior in the novelty-investigation test (Supplementary Fig. 2). Next, we tested long-term memory using the

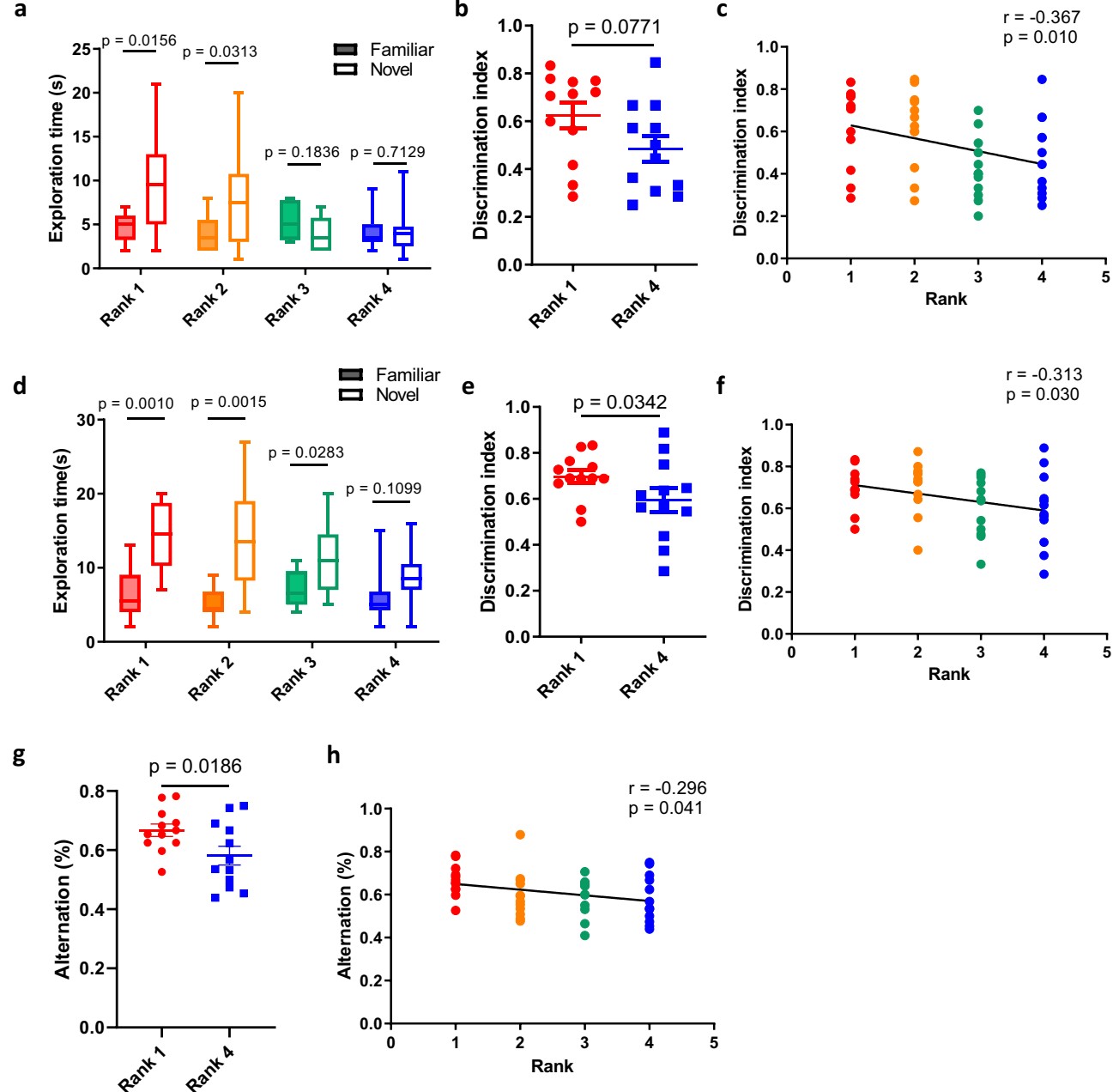

**Fig. 1 Weanling mice with a higher rank showed better memory performance. a** Exploration time to familiar and novel objects in weanling mice with different social ranks in the 1-hour NOR test (Wilcoxon signed-rank test, $n = 12$ cages). **b** Discrimination index between 1st- and 4th-rank weanling mice in the 1-hour NOR test (Wilcoxon signed-rank test, $n = 12$ pairs). **c** The correlation between social ranks of weanling mice and the discrimination index in the 1-hour NOR test (Spearman correlation, $n = 12$ cages). **d** Exploration time to familiar and novel objects in weanling mice with different social ranks in the 24-hour NOR test (Wilcoxon signed-rank test, $n = 12$ cages). **e** Discrimination index between 1st- and 4th-rank weanling mice in the 24-hour NOR test (Wilcoxon signed-rank test, $n = 12$ pairs). **f** The correlation between social ranks of weanling mice and the discrimination index in the 24-hour NOR test (Spearman correlation, $n = 12$ cages). **g** The spontaneous alternation rate between 1st- and 4th-rank weanling mice in the Y maze (Wilcoxon signed-rank test, $n = 12$ pairs). **h** The correlation between social ranks of weanling mice and the spontaneous alternation rate in the Y maze (Spearman correlation, $n = 12$ cages). The boxplots show the minimum, 25th percentile, median, 75th percentile, and maximum values. Error bars = SEM.

NOR test with a 24-hour interval between the training and test sessions. Similar to the results for short-term memory, there was no difference in the exploration time between the familiar and novel objects in 4th-rank weanling mice (Fig. 1d). The discrimination index was significantly different between 1st- and 4th-rank mice and negatively correlated with social rank (Fig. 1e, f). To further explore the correlation between social rank and memory ability, we also applied the spontaneous-alternation Y maze to examine spatial working memory in weanling mice[46].

The results showed a significantly higher alternation rate in 1st-rank mice than in 4th-rank mice (Fig. 1g). The negative correlation between alternation rate and social rank was also significant (Fig. 1h), again indicating better memory in dominant mice than in subordinate mice. Together, these results suggested that weanling mice with a higher rank generally had better short- and long-term recognition memory as well as spatial memory.

Since weanling mice at 3 weeks of age are still at the developmental stage, it is possible that the differences in social

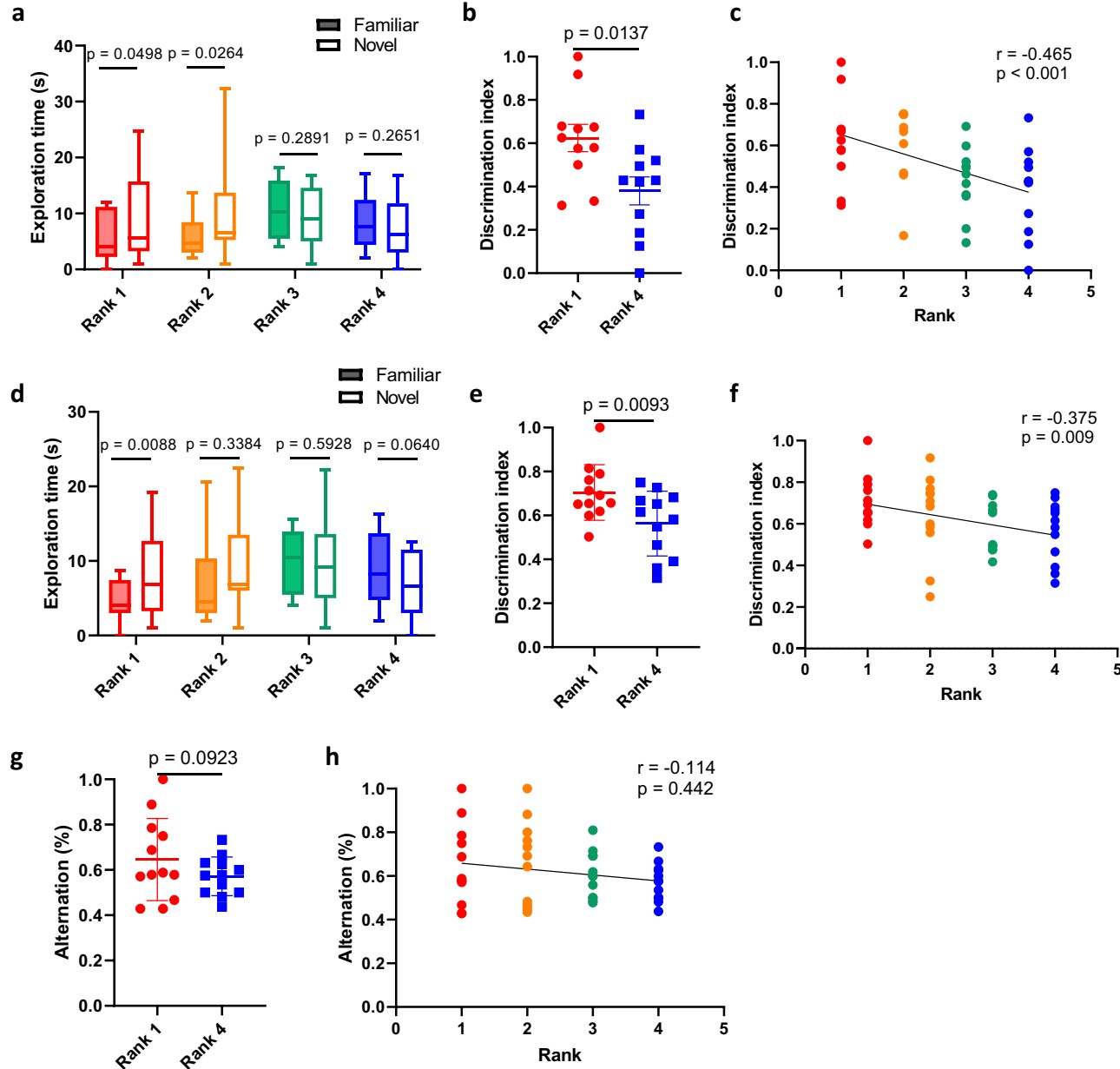

**Fig. 2 Adult mice with a higher rank showed better memory performance. a** Exploration time to familiar and novel objects in adult mice with different social ranks in the 1-hour NOR test (Wilcoxon signed-rank test, $n = 12$ cages). **b** Discrimination index in the 1-hour NOR test between 1st- and 4th-rank adult mice (Wilcoxon signed-rank test, $n = 12$ pairs). **c** The correlation between social ranks of adult mice and the discrimination index in the 1-hour NOR test (Spearman correlation, $n = 12$ cages). **d** Exploration time to familiar and novel objects in adult mice with different social ranks in the 24-hour NOR test (Wilcoxon signed-rank test, $n = 12$ cages). **e** Discrimination index in the 24-hour NOR test between 1st- and 4th-rank adult mice (Wilcoxon signed-rank test, $n = 12$ pairs). **f** The correlation between social ranks of adult mice and the discrimination index in the 24-hour NOR test (Spearman correlation, $n = 12$ cages). **g** The spontaneous alternation rate between 1st- and 4th-rank adult mice in the Y maze (Wilcoxon signed-rank test, $n = 12$ pairs). **h** The correlation between social ranks of adult mice and the spontaneous alternation rate in the Y maze (Spearman correlation, n = 12 cages). The boxplots show the minimum, 25th percentile, median, 75th percentile, and maximum values. Error bars = SEM.

ranks and memory ability are due to different levels of maturity. However, in 8-week-old adult mice, we still observed better performance for both short-term and long-term memory in dominant mice than in subordinate mice (Fig. 2a–f), suggesting that the difference in object-recognition memory was not simply caused by developmental maturity. For the spontaneous-alternation Y maze, we also observed that dominant mice tended to have better spatial memory, although the data were not statistically significant (Fig. 2g, h).

**There was higher LTP in hippocampal neurons in 1st-rank mice than in 4th-rank mice.** Hippocampal LTP is known to be critical for memory formation and memory consolidation[47]. To determine whether the enhanced memory ability in mice with a higher social rank is associated with augmented LTP, we measured the field excitatory postsynaptic potentials (fEPSPs) on the dendritic fields of CA1 neurons and stimulated CA3–CA1 Schaffer collaterals in acute hippocampal slices prepared from either 1st- or 4th-rank weanling mice. Our data showed that acute

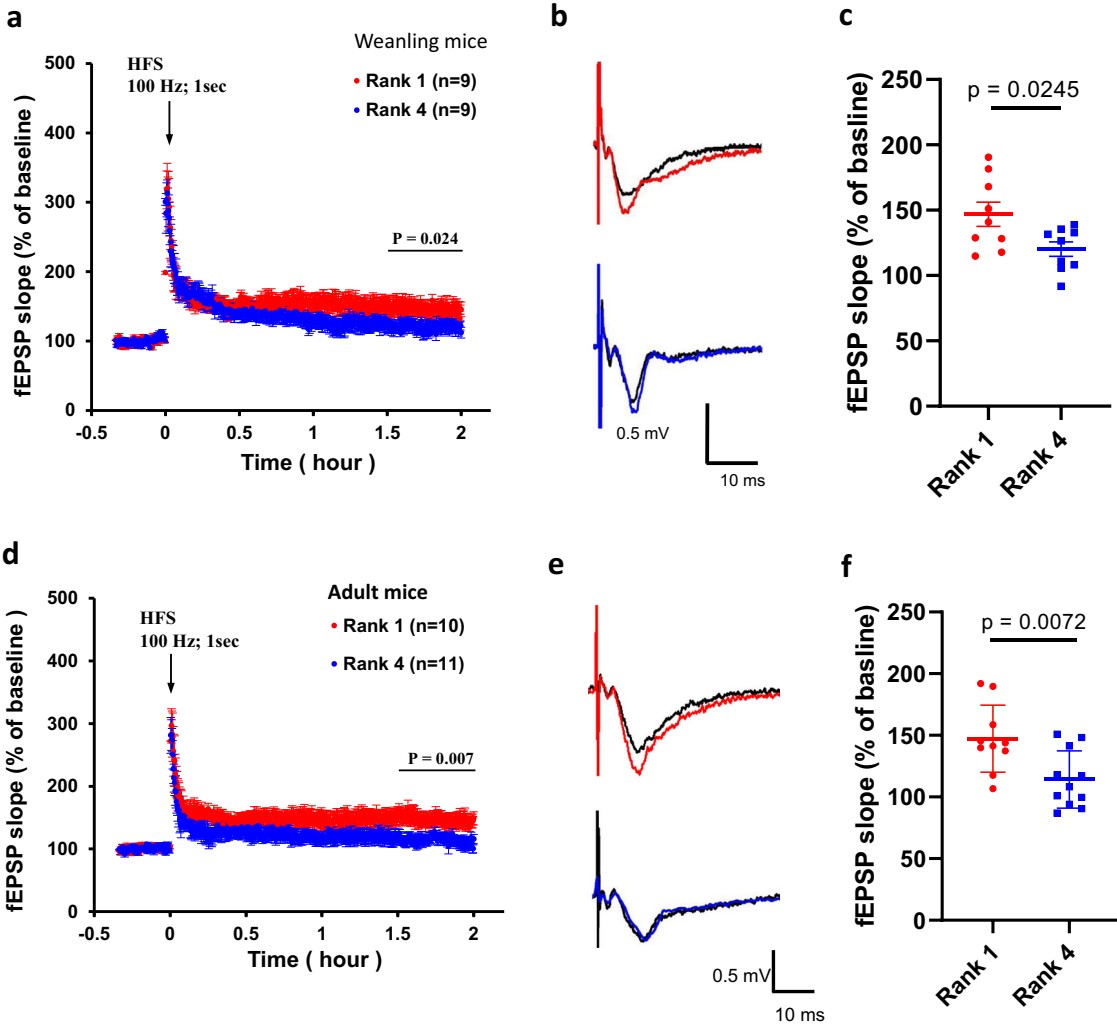

**Fig. 3 Higher-rank mice showed greater LTP in hippocampal slices. a** Augmented LTP induced by a one-second high-frequency stimulation (100 Hz) in hippocampal slices of 1st- and 4th-rank weanling mice. **b** Representative fEPSPs before (black) and after (red: 1st-rank mouse; blue: 4th-rank mouse) LTP induction in weaning mice. **c** fEPSP slopes measured 2 h after LTP induction in 1st- and 4th-rank weaning mice (unpaired *t* test, *n* = 9 pairs). **d** Augmented LTP induced by a one-second high-frequency stimulation (100 Hz) in hippocampal slices of 1st- and 4th-rank adult mice. **e** Representative fEPSPs before (black) and after (red: 1st-rank mouse; blue: 4th-rank mouse) LTP induction in adult mice. **f** fEPSP slopes in 1st- and 4th-rank adult mice (unpaired *t*-test, *n* = 10–11 pairs). Error bars = SEM.

hippocampal slices prepared from 1st-rank mice had greater LTP than slices isolated from 4th-rank mice (Fig. 3a–c). A similar phenomenon was also shown in slices prepared from adult mice (Fig. 3d–f). The results therefore supported positive correlations between social rank and memory.

**Memory-improving drugs enhanced social status**. Several genes have been shown to modulate mouse hippocampal LTP as well as learning and memory in mice[48]. The identification of greater LTP in high-rank mice led us to measure the abundance of these memory-related genes in the hippocampus of weanling mice. The results of quantitative PCR showed that social rank was positively correlated with the expression levels of *Grin2b* (NR2B)[49], a subunit of the NMDA receptor that improves synaptic plasticity and memory in neurons, and *Phf2*, a histone demethylase (Supplementary Fig. 3a, b). *Grin2B* and *Phf2* have been suggested to be involved in the BDNF/TrkB/CREB signaling pathway for memory consolidation[48]. Although the data were not statistically significant, dominant mice tended to have a higher expression of genes involved in this pathway, including *Bdnf*, *Ntrk2* (TrkB), *Creb*, *Cdk5*, and *Camk2* (Supplementary Fig. 3). We therefore

further examined the consistency between social rank and expression level of each gene in 6 pairwise comparisons. Generally, mice with higher social ranks were more likely to have higher expression, especially of *Phf2*, *Creb*, *Grin2b*, and *GluR1* (Fig. 4a and Supplementary Table 1). The expression of *CamK2* also showed a similar tendency (*p* = 0.053). It is noteworthy that these consistencies were more robust in comparison with larger-rank distances, e.g., between the 1st and 4th rank, than smaller-rank distances, e.g., between the 3rd and 4th rank.

The CREB signaling pathway can be activated by inhibition of histone deacetylase (HDAC)[50]. It has been shown that SB, a HDAC inhibitor, can induce hippocampal LTP and enhance memory[51]. We therefore tested the effect of sodium butyrate on the dominant behavior of mice. As expected, weanling mice injected with SB intraperitoneally for 2 weeks showed a higher discrimination index in the NOR test than control mice (Fig. 4b). Administration of SB also increased the mouse winning rate and social status in the tube test (Fig. 4c, d). CREB signaling can also be activated by a phosphodiesterase inhibitor, rolipram, which has been shown to facilitate LTP and improve memory[52]. After 2 weeks of treatment, the weanling mice showed a better

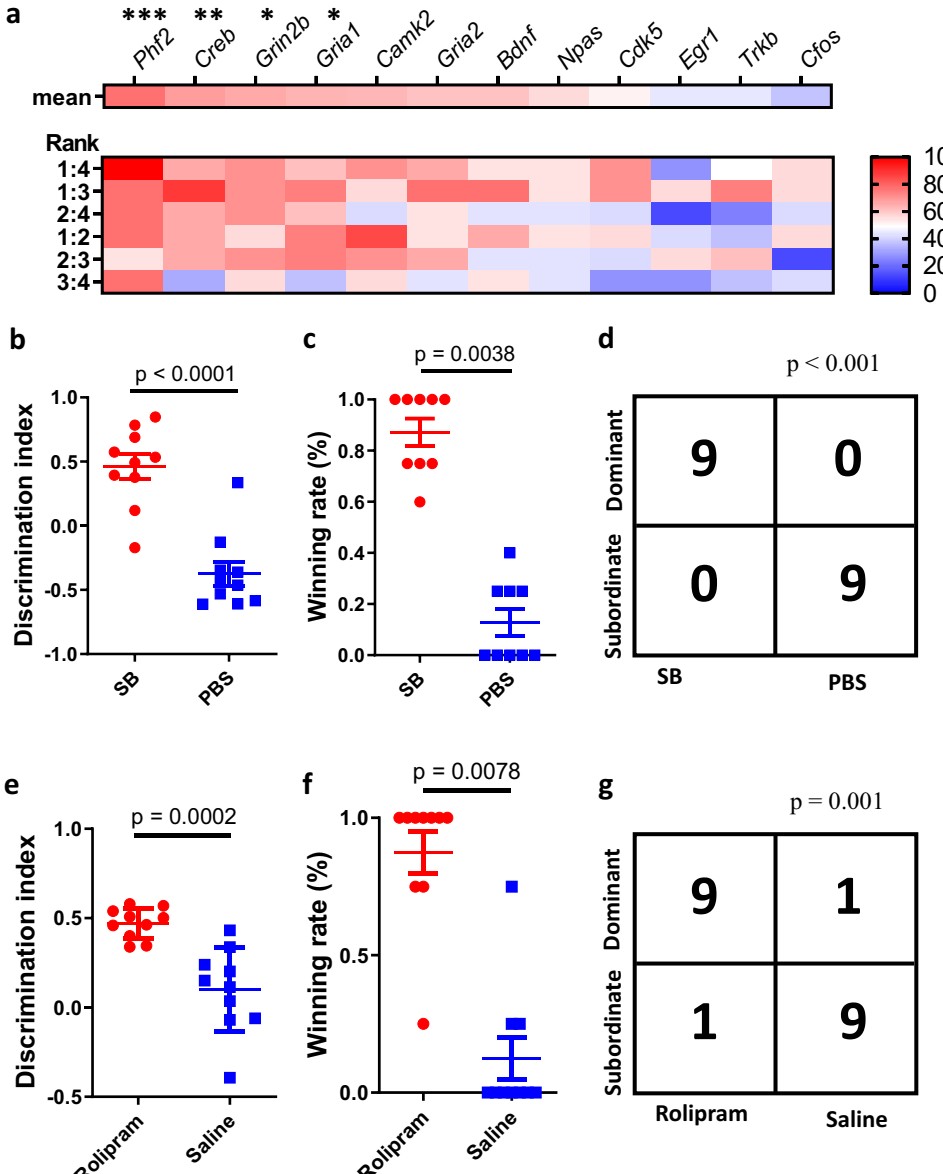

**Fig. 4 Memory-improving drug-enhanced social dominance. a** The consistency between social rank and expression levels of memory-related genes in 6 pairwise comparisons. The value in the colored font indicates the percentage of higher-rank mice with a higher gene expression than lower-rank mice (binary rank-correlation test, $n = 7$–9 cages). **b** Discrimination index in the 1-hour NOR test between SB-treated mice and PBS-treated mice (unpaired $t$-test, $n = 10$ for each group). **c** Winning rate in the tube test between SB-treated mice and PBS-treated mice (Wilcoxon signed-rank test, $n = 9$ for each group). **d** The contingency table for the relationship between social dominance and SB treatment (Fisher's exact test, $n = 18$). **e** Discrimination index in the 1-hour NOR test between rolipram-treated mice and saline-treated mice (unpaired $t$-test, $n = 10$ for each group). **f** Winning rate in the tube test between rolipram-treated mice and saline-treated mice (Wilcoxon signed-rank test, $n = 10$ for each group). **g** The contingency table for the relationship between social dominance and rolipram treatment (Fisher's exact test, $n = 20$). Error bars = SEM.

discrimination index along with an increase in dominant behavior and social status (Fig. 4e–g). Together, our results based on SB and rolipram suggested that improving memory pharmacologically can also enhance social dominance.

**High-rank children exhibited better memory ability.** The discovery in weanling mice raised a question of whether the correlation between social rank and memory could also be observed in young human children. To answer this question, we conducted behavioral tests in 164 preschool-aged children. Based on the experimental design in the mouse tube test, we first arranged the young children into groups of four and established their social hierarchies by evaluating their behaviors while they played a

competitive bunny game with a round-robin design. The bunny game was used as a comparable task to the mouse tube test in a previous study[32]. We next used the Picture memory subtest (for recognition memory) and Zoo subtest (for spatial working memory) from the Wechsler Preschool and Primary Scale of Intelligence through one-by-one testing to compare memory ability between 1st- and 4th-rank children ($n = 82$) (Fig. 5a)[53]. For both assays, 1st-rank children tended to perform better than 4th-rank children (Supplementary Fig. 4a, b). The working-memory index (WMI), which integrated two memory subtests, showed a significantly higher score in 1st-rank children than in 4th-rank children (Fig. 5b). In addition to the bunny-game evaluation conducted in a well-designed experimental context, we also asked the preschool teachers to

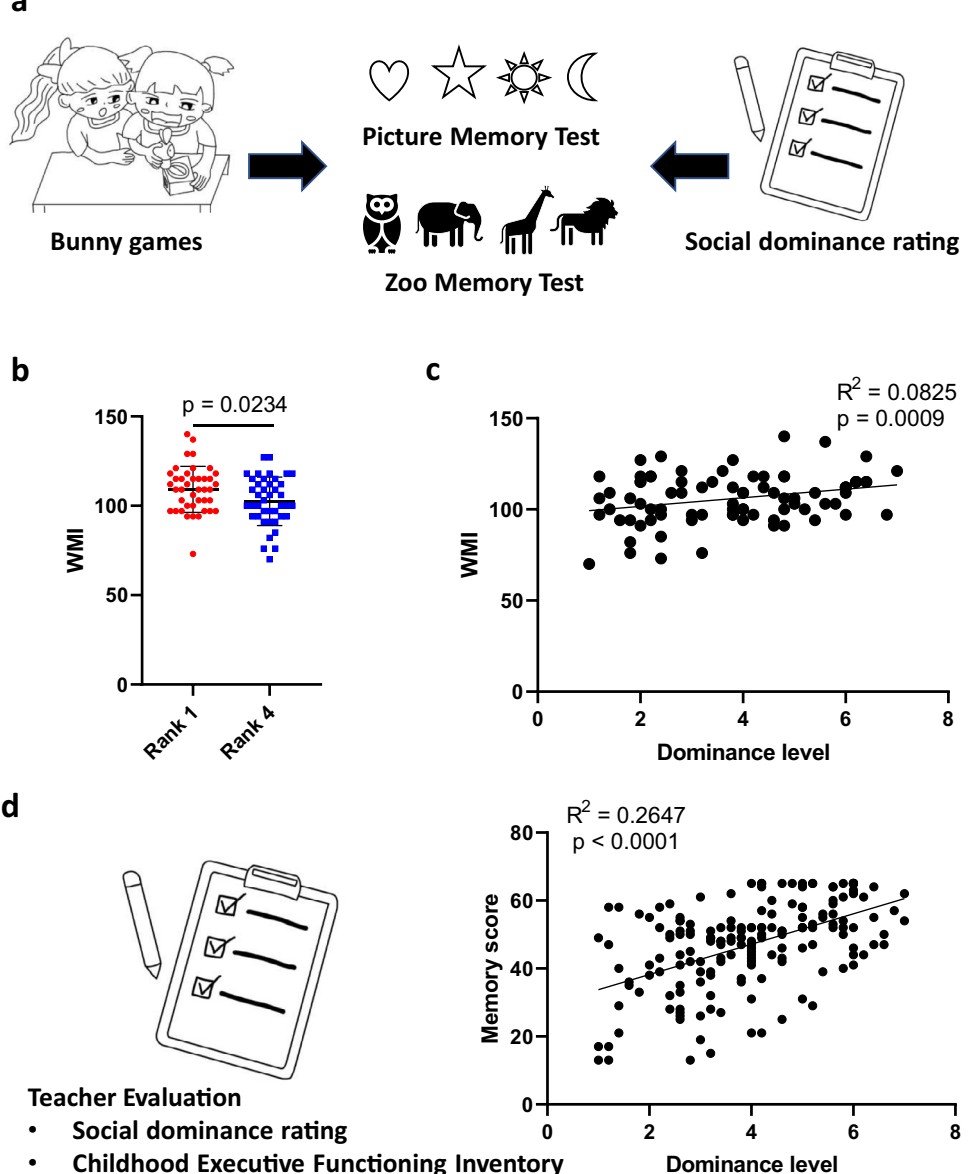

**Fig. 5 Children with a higher rank or dominance levels showed better memory ability. a** Social ranks were defined by bunny games. The dominance levels were evaluated by social-dominance rating. The working-memory index (WMI) was evaluated and integrated by picture-memory and zoo-memory tests. The image is adapted from Chou et al. (2021)[32]. **b** The WMI for 1st- and 4th-rank children (unpaired t-test, n = 82). **c** The correlation between the WMI and dominance levels (Pearson correlation, n = 82). **d** The correlation between memory score evaluated by the Childhood Executive Functioning Inventory and dominance level evaluated by social-dominance rating (Pearson correlation, n = 175). Error bars = SEM.

evaluate the children's social-dominance level according to their daily observations using the Social Dominance Rating Scale (Fig. 5a), which has been shown to be consistent with the social ranks determined by the bunny game[32,54]. The dominance levels in these 82 children showed trends correlated with memory scores in both Picture memory and Zoo subtests (Supplementary Fig. 4c, d). The correlation between the WMI and dominance levels was also statistically significant (Fig. 5c).

To further validate the finding of better memory in dominant children, another 175 young children from 3 other preschools were recruited for the inventory study (Fig. 5d). The Working Memory subscale of the Childhood Executive Functioning Inventory was used to evaluate the children's working-memory performance based on teachers' observations in preschool classrooms and was thought to reflect the children's abilities to remember classroom rules and to retrieve learning experiences

from long-term memory for application in daily interactions[55]. Consistent with previous data based on the Wechsler scale, the memory score evaluated by the Childhood Executive Functioning Inventory was also correlated with dominance levels (Fig. 5d). Together, the results from both behavioral tasks and inventories suggested that memory ability in children was positively correlated with social status as well as dominance levels.

**Memory ability was correlated with prosocial-strategy use but not coercive-strategy use.** To investigate the potential functions of memory in the formation of social hierarchy, for 175 children in the inventory study, we applied the Resource Control Strategy Scale to measure children's resource-control ability and strategy usage[56]. As expected, children with higher social-dominance levels possessed better abilities in resource-control and adopted more prosocial strategies and coercive strategies (Fig. 6a–c).

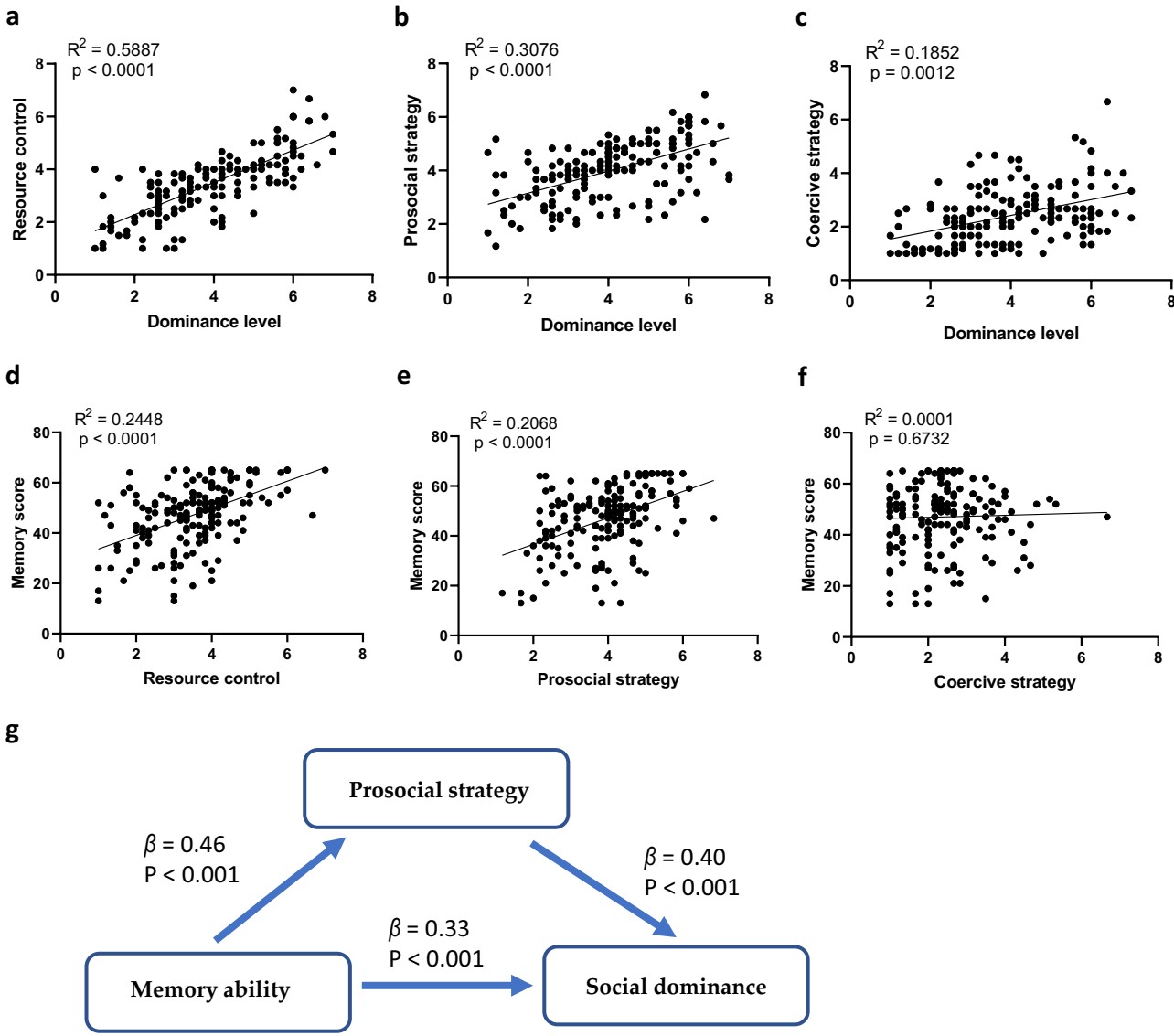

**Fig. 6 Memory ability was correlated with prosocial strategy use but not coercive strategy use. a** The correlation between dominance level and resource control (Pearson correlation, $n = 175$). **b** The correlation between dominance level and prosocial strategy (Pearson correlation, $n = 175$). **c** The correlation between dominance level and coercive strategy (Pearson correlation, $n = 175$). **d** The correlation between memory score and resource control (Pearson correlation, $n = 175$). **e** The correlation between memory score and prosocial strategy (Pearson correlation, $n = 175$). **f** The correlation between memory score and coercive strategy (Pearson correlation, $n = 175$). **g** The mediated model linking children's memory ability and children's social dominance (regression analysis, $n = 175$).

However, the memory score was surprisingly only related to resource control with children's use of prosocial strategies (Fig. 6d, e) but not coercive strategies (Fig. 6f). These results suggested that memory may play a crucial role in the acquisition and adoption of prosocial strategies, which then improve the resource-control ability. We therefore further asked whether the relationship between children's memory and social status is mainly due to the mediation of prosocial-strategy learning. The regression analysis of the mediation effect showed that all $\beta$-values in the four regression analysis steps reached significant levels (Supplementary Table 2). According to Baron Kenny's (1986) criterion[57], this result demonstrated a "partial mediation effect". The Sobel test was also conducted with a 95% confidence interval and showed a significant effect (Sobel $z = 5.221$, $p < 0.001$). The results therefore indicated a mediation effect of prosocial strategy and supported our proposed model that memory could help children learn more adaptive social strategies, which further enhance their social dominance (Fig. 6g).

**Children with a higher social rank were superior in processing facial cues of social dominance**. The regression analysis also suggested that there is a direct relationship between memory and social dominance. To explore the direct function of memory on social-dominance formation, we designed an event-related potential (ERP) study to investigate how social status affects children's processing of social-dominance cues. In the task, 1st- and 4th-rank children based on the bunny game ($n = 24$) were asked to watch peer faces with different social facial expressions (dominant, neutral, and subordinate faces) on the screen (Fig. 7a)[58–60]. To remove other irrelevant effects and ensure that the experimental stimulus specifically triggered the processing of social-dominance information, all the facial photographs were not repeated to avoid the face-recognition effect, and the presentation order of the facial-expression types was randomly assigned to avoid the old/new effect.

Our ERP study focused on FN400, a negative component elicited around 300–450 ms at the frontal areas[61,62], which is related to

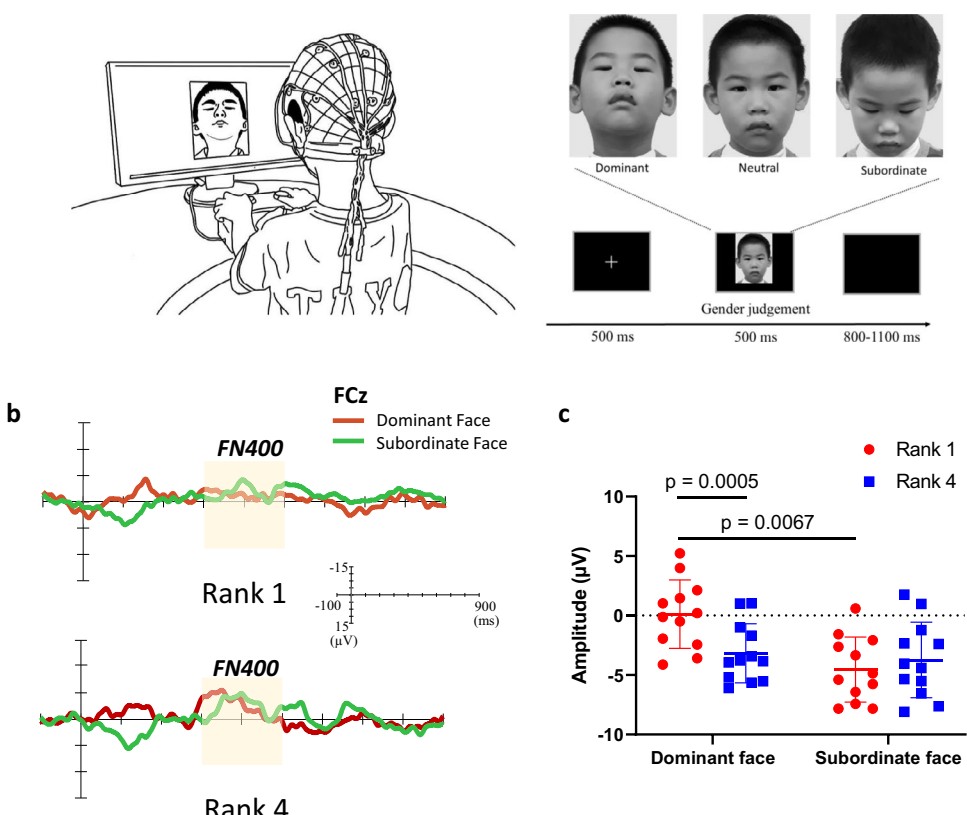

**Fig. 7 Children with a higher rank were more capable of processing dominant faces. a** Experimental procedure and stimuli of the ERP study. EEG recordings were acquired from preschool children while they watched photographs of the faces of their peers (with representative images for dominant, neutral, and subordinate facial expressions). The photograph of the child is used with the consent of the parents. **b** Grand-mean ERP difference waveforms of FN400 in the FCz channel in response to dominant and subordinate faces in 1st- and 4th-rank children (n = 12 pairs). **c** The difference wave amplitudes of FN400 in response to dominant and subordinate faces in 1st- and 4th-rank children (two-way ANOVA, n = 12 pairs). *Note*. The FN400 amplitudes of each facial expression were corrected by subtracting the averaged amplitude of the neutral facial condition. Error bars = SEM.

implicit memory during semantic category-based inferences[63–65]. By employing difference waves to exclude individual difference (via subtracting the averaged amplitude of the neutral-face condition) (Fig. 7b), the two-way analysis of variance (ANOVA) of FN400 showed that the main effects of both social status and social-dominance facial expressions, as well as their interaction, were significant (Supplementary Table 3). In the between-group comparison, when processing the dominant facial expression, 1st-rank children displayed a smaller FN400 amplitude than 4th-rank children (Fig. 7c and Supplementary Table 4). In the within-group comparison, 1st-rank children exhibited a smaller FN400 amplitude in response to dominant faces than to subordinate faces, in contrast, such a difference in FN400 amplitude was not found in 4th-rank children. Since a larger FN400 amplitude is evoked by the processing of unexpected fluency compared with previous social knowledge[66], a smaller FN400 amplitude in response to dominant faces implies that 1st-rank children have a better fluent ability in processing dominant faces than 4th-rank children. The results of our ERP study, therefore, implied that children with a higher social rank have superior implicit memory in recognizing social-dominance cues.

## Discussion

In this study, by examining the relationships of social hierarchy with memory abilities in mice, we found that dominant mice showed better performance in recognition and spatial working memory, along with augmented LTP, and higher expression of memory-related genes in the hippocampus. The relationship between social hierarchy and memory was further strengthened by the dominant status of mice treated with memory-improving drugs. To explore this phenomenon across species, we combined multiple approaches to demonstrate that children with a higher social status also have better memory ability. The data further suggested that better memory may assist children in acquiring prosocial strategies and recognizing social-dominance cues. We believe this is the first report to present an association between social hierarchy and memory in two species in parallel, especially in humans. While our mouse model indicated the possible molecular and neural mechanisms underlying the association, the study in human children revealed the functional importance of memory in acquiring and maintaining dominance status. By utilizing different species to answer specific questions and reveal information that could not be approached solely by one species, our study not only presented a remarkable similarity between children and mice, but also provided important information to the research of social interaction, as well as of learning and memory in the fields of biology, psychology, and education.

Animals in the wild are constantly facing changing environments. Better memory could therefore enhance individuals' opportunities to access resources and potentially obtain a higher

social status. Although dominance with better spatial learning has been implicated in a few studies[25,67,68], recent research in mice failed to detect any association between social rank and memory[43,44]. In contrast, our data consistently demonstrated this relationship in short- and long-term memory, in young and old mice, and in recognition (NOR) and spatial working memory (Y maze). The greater hippocampal LTP and higher expression of memory-related genes in dominant mice further supported this association. More importantly, daily administration of SB or rolipram, both of which activate CREB signaling to improve memory ability, also enhanced mouse dominant behaviors in the tube test. However, although the function of the hippocampus in recognition memory has been reported in numerous studies[69–74], its role in the NOR assay remains controversial[75]. Whether other brain regions critical for recognition memory, such as the perirhinal cortex[70,74], are also involved in social-hierarchy formation, may warrant further study.

The findings in mice led us to identify better memory in dominant children, which, to our knowledge, has never been reported in previous research. Children with higher social ranks also showed better short- and long-term memory, recognition memory, spatial working memory, and perhaps implicit memory. Although the correlations between social hierarchy and memory are extremely similar to those in mice, the neural mechanisms underlying this relationship in humans remain to be investigated. Unfortunately, hippocampal activity in humans cannot be easily assessed by electroencephalography. Quantification of gene expression in the human hippocampus is even more challenging. However, previous studies have reported that there are sequence variations in *Gria1*, *Grin2B*, and *Creb* in humans[76–79]. The polymorphisms of *Grin2B* have also been shown to be correlated with memory ability[76,77]. Whether these variants are related to social behaviors in humans remains to be explored.

Identifying social status associated with memory provided inspiring insights into human children's social interactions. Our previous research found that the formation of social hierarchy is affected by intrinsic characteristics[32]. The differences in temperament between individuals make the early formation of social hierarchy easier and less conflicting. However, young children still need to learn to interact with their peers while the social hierarchy is initially formed. The present study revealed that social-dominance level and resource-control ability were strongly correlated, and that better memory was related to an increased adoption of prosocial strategies. To compete for resources and status in the classroom, children may instinctively use coercive strategies, which are usually prohibited by teachers and ostracized by peers. The flexible adoption of prosocial strategies is obviously more acceptable and effective during social interaction[9,12]. Previous studies have shown that the use of prosocial strategies could be influenced by parenting or school moral education, or could be due to the accumulation of interaction experience. However, the reason why some children can effectively learn and flexibly use these "smarter" prosocial strategies, while other children exposed to the same school instruction cannot, has not been well investigated. Our studies based on both behavioral tasks and inventories suggested an advantage of better memory in learning and using prosocial strategies. Through reciprocal exchange or acting cooperation, regardless of whether their behavior is motivated by altruism, dominant children can enhance their opportunities to obtain more resources and maintain their position in a group.

Although memory could affect social status indirectly through learning social strategy, the regression analysis of the mediation effect also suggested a direct influence of memory on social status. ERP studies focusing on FN400 implied another advantage of memory in processing socially dominant facial expressions. FN400 has been proposed to reflect the familiarity of recognition and has been treated as an indicator of conceptual fluency/priming across different stimuli[61–63,65]. The amplitude of FN400 is usually negatively correlated with the coherence/homogeneity of the conceptual category in the brain[64]. Based on this hypothesis, a smaller FN400 amplitude in high-rank children in response to a dominant face possibly reflects better fluency in recognizing signs of social dominance. Social status significantly correlated with different abilities in processing social expression, particularly better implicit memory pertinent to dominant facial information in high-rank children. Previous observational studies have shown that preschool children used coercive strategies to compete for resources at the beginning of a new semester, but gradually incorporated prosocial strategies[4,9]. Notably, once the social hierarchy was initially established, high-status children did not utilize prosocial strategies all the time, instead, they applied coercive strategies while interacting with low-status children, but flexibly adopted prosocial strategies or even dual strategies when encountering high-status children[9,56,80]. Our ERP study further showed that high-rank children had better implicit memory of dominant cues, suggesting that identifying individuals who are possible opponents that need to be handled more carefully and determining effective strategies could be the keys to achieving a higher social status.

Finally, the causality between social hierarchy and memory ability remains to be investigated. Whereas better memory could potentially help animals learn new strategies to acquire social dominance, such as prosocial strategies in human children, it is also possible that memory ability is modulated by social status. For example, it has been suggested that subordinate animals constantly experience chronic social stress[21], which could affect learning and memory through multiple mechanisms, including modulation of neuronal activity, gene expression, and neurogenesis in the hippocampus[81]. Similarly, stressful environments, such as those associated with reduced social interaction, less attention from parents/teachers/peers, or limited availability of resources, could also affect children's memory ability. In the future, an intervention design to further evaluate mice and a cross-lagged longitudinal study to assess children would be helpful for us to answer this chicken-and-egg question between social hierarchy and memory.

In summary, our study revealed a positive correlation between social hierarchy and memory ability in mice and children. We believe that better memory is advantageous for young children, both in acquiring advanced dominance strategies and detecting social dominance cues to achieve and maintain a higher social status. The abundant resources and ample interaction opportunities accompanied by a higher social status potentially further create a favorable environment for memory development for these children. This finding has important implications for child education. Most children want to be the center of the group, hope that their opinions can be taken seriously, and desire to play with the toys they want. According to the findings of this study, however, social-strategy use and resource-control ability are largely influenced by learning and memory. To assist young children in achieving better social adaptation, we cannot just provide moral education or behavioral requirements. Instead, we should take the level and limitations of children's cognitive development into account. It is difficult for those who are not yet able to learn better social strategies or are not yet mature in detecting social cues to strive for social status or obtain classroom resources. The relatively few resources and poor interaction quality caused by a low social status may in turn lead to frustration, anxiety, and fewer opportunities to encounter cognitive stimuli, consequently leading to poor memory development. Therefore, parents and teachers should not just punish children when they use improper strategies, but should also set class rules to ensure that every child

has the opportunity to be listened to and to play with the toys they want. For children with better memory and learning abilities, we should pay attention to their motives underlying their seemingly prosocial behaviors. Through more discussion and guidance and by leading them to concern more about others' needs, parents and educators can help children develop genuine altruism.

## Methods

**Mice.** C57BL/6 J adult male mice between the ages of 8 and 10 weeks and weanling mice between the ages of 3 and 4 weeks were purchased from the National Laboratory Animal Center in Taiwan. Mice were housed in a cage of four in a controlled animal room with a 12-h light/dark cycle (0700–1900 h). All tests were conducted during the light period, and all animal procedures were in compliance with institutional guidelines established and approved by the Institutional Animal Care and Use Committee of National Tsing Hua University and Academia Sinica.

### Behavioral assay for mice

*Tube test.* The tube test was based on a previous study[32]. The assay was modified from the standard tube test[82], but the mice were not trained first. A clear Plexiglas tube (3.75 cm diameter, 60 cm length) was used for adult mice, and a narrower tube (2.5 cm diameter, 60 cm length) was used for weanling mice. Mice were habituated to the procedure room for 1 h on two consecutive days. On the third day, a tube-test trial was carried out involving two mice that were simultaneously released at opposite ends of the tube and then ran toward the middle. When a mouse retreated and set all four paws outside the tube, the test trial was over and that mouse was considered the loser.

To establish social ranks among four mice in a cage, a round-robin design was applied to the four mice housed in each cage to allow the six possible pairs to compete. For each pair, mice were tested against one another in this manner in four consecutive trials, with each mouse starting at an alternative end of the tube for each trial. The interior of the tube was cleaned after every pairwise with 70% ethanol. The final social ranks were based on four trials between two individuals. If the winning numbers between two individuals were equal, i.e., 2 and 2, the rank was determined by the total number of wins for each animal across all comparisons. For the comparison between drug-treated mice and control mice, the winning rate between two mice was calculated as the percentage of the number of winnings in four trials.

*Novel object recognition.* The novel object-recognition test was performed as described previously[45]. The experiment was divided into familiar and testing phases. In the familiar phase, each mouse was placed at the edge of an empty open cage [28 cm (L) × 16.5 cm (W) × 13 cm (H)] containing two identical plastic blockers at the bottom of the cage. Mice were placed into the cage to freely explore the cage for 10 min to familiarize the blockers, then returned to the homecage for 1 h (short-term memory) or 24 h (long-term memory). In the testing phase, each mouse was moved back to the same open cage for 5 min with one familiar blocker used in the familiar phase and one different blocker with a contrasting color and shape. The total time for which the mice spent exploring the blockers (nose touching) was recorded. The apparatus was cleaned with 70% ethanol between each trial. The exploration-time difference between the familiar object and novel object divided by the total exploration time was calculated as the discrimination index by individuals.

*Novelty-investigation test.* The novelty-investigation test was modified from the novel object-recognition test[45]. A mouse was introduced into a box with one unfamiliar object in the center of an empty open cage [28 cm × 16.5 cm × 13 cm] for free exploration for 10 min. The object-investigation time was recorded to represent exploration activity. The apparatus was cleaned with 70% ethanol between each trial.

*Spontaneous alternative Y maze.* The experimental design was based on a previous study[46]. The Y maze consists of three white, opaque arms [30 cm (L) × 7 cm (W) × 16 cm (H)] at a 120° angle from each other. Each mouse was placed at the center of the maze and allowed to freely explore the maze for 5 min. The behavior of mice in the Y maze was recorded by a camera and evaluated by SMART VIDEO TRACKING Software (Panlab) to obtain the number of entries into each arm. The number of arm entries was used to calculate the alternative rate. The apparatus was cleaned with 70% alcohol and air-dried between each mouse.

### Electrophysiology of mouse hippocampal slices

Mice at 4 or 8 weeks of age were first anesthetized by isoflurane (Panions & BF Biotech Inc.) and then sacrificed by decapitation. The brain was removed rapidly, and the hippocampi were dissected out in icy-cold artificial cerebrospinal fluid (aCSF) containing (in mM) 119 NaCl, 2.5 KCl, 1 NaH$_2$PO$_4$, 1.3 MgSO$_4$, 26.2 NaHCO$_3$, 2.5 CaCl$_2$, and 11 D-glucose, and oxygenated with 95% O$_2$/5% CO$_2$. Hippocampal slices (300 μm thick) were cut along the long axis using a 5100-MHz vibratome (Campden Instruments

Ltd.) in the icy-cold aCSF. Then, the slices were first recovered in aCSF at 34 °C for 30 min and switched to room temperature for at least 2 h. For field recording, a slice was transferred to the recording chamber and continuously perfused with aCSF (1 ml/min) at room temperature. The afferent input from CA3 was severed by making a cut between the CA1 and CA3. The recording electrodes were pulled from borosilicate-glass capillary tubes (1.5-mm outer diameter, 0.86-mm inner diameter, World Precision Instruments) using a single-stage glass microelectrode puller (PP-830, Narishige) and filled with 3 M NaCl. The field excitatory post-synaptic potentials (fEPSPs) were evoked by placing a bipolar tungsten electrode (A-M SYSTEMS) on the Schaffer collateral/commissural pathway. An Axon200B amplifier (Molecular Devices Corp.) together with Digidata 1440 was used for data acquisition. Data were filtered at 1 kHz and sampled at 25 kHz with Clampex 10.7 software (Molecular Devices Corp.). The stimulation strength was varied between 10 and 100 μA, and the intensity that elicited 50% of the maximum response was chosen for the LTP experiments. The stimulation pulse was delivered every 15 s. Once a 20-min-long stable baseline was achieved, a train of 100-Hz stimulation was delivered for 1 s to induce LTP, followed by fEPSP recording every 15 s for 2 h.

### RNA extraction, cDNA synthesis, and quantitative real-time PCR

After identifying social ranks, mice were sacrificed for isolation of the hippocampus. Total RNA was extracted using RNeasy Mini Kit (Qiagen), including DNase (Qiagen) treatment. cDNA of mRNA was generated by FIREScript® RT cDNA Synthesis Mix (Solis Biodyne) using oligo dT as the primer. The qPCR reactions were performed under a qTOWER³ real-time PCR system (Analytik Jena) using 5x HOT FIREPol® EvaGreen® qPCR Mix Plus (Solis Biodyne) following the manufacturer's protocol. The relative expression was calculated using the ΔCT method, and *GAPDH* was used as a normalization control. The primer sequences used for qPCR are shown in Table S5.

### Drug administration

*Sodium butyrate.* Sodium butyrate (Sigma, 156-54-7) was dissolved in 0.01 M PBS and administered intraperitoneally (i.p.) daily for two weeks at a dose of 1.2 g/kg. Control animals received an injection of the same volume of vehicle (PBS).

*Rolipram.* Rolipram (HelloBio, 61413-54-5) was first dissolved in 2% DMSO and then 0.9% sterile saline solution and administered intraperitoneally (i.p.) daily for two weeks at a dose of 1 mg/kg. Control animals received an injection of the same volume of vehicle (sterile saline).

**Children participants.** The study participants were recruited from five preschools in Hsinchu City, Taiwan. For the behavioral studies, 164 children from two preschools (88 boys and 76 girls, average age, 67.18 ± 9.09 months) participated in a competitive bunny game to establish social ranks. The memory abilities of the 1st- and 4th-rank children among the 164 participants were further tested individually by the *Wechsler Preschool and Primary Scale of Intelligence*. The dominance-level and resource control abilities in 1st- and 4th-rank children were also evaluated by the *Social Dominance Rating Scale* and *Resource Control Strategy Scale*. For the inventory studies, another 175 children from three of the other preschools (102 boys and 73 girls, average age, 66.52 ± 8.33 months) were evaluated by the *Social Dominance Rating Scale*, *Childhood Executive Functioning Inventory*, and *Resource Control Strategy Scale*. For the ERP study, 12 children each in the 1st and 4th ranks participated in the electroencephalogram (EEG) data collection. Consent forms were provided by parents for all children in this study (approved by the National Tsing Hua University Research Ethics Committee 10804HT020).

### Behavioral studies in children

*Bunny game.* The social ranks of 164 children were evaluated by the competitive bunny game based on a previous study[32]. Briefly, children were randomly divided into groups of four. Paired encounters were staged with a round-robin design (comparable to the design in the mouse study), such that each child would encounter every other child in the same group, leading to a total of six combinations of pairs. For each round of the bunny game, a tester presented a picture card showing the required placement of three wooden blocks, colored red, blue, and yellow. The children were told to place the blocks in the correct positions before putting the rabbit into the hole representing its home. The first child to place his or her bunny doll into its hole was declared the winner. All games were conducted at preschool before lunch time (9 am–11 am).

*Memory tests.* After one month of the bunny game, the two subscales of the Chinese version of the Wechsler Preschool and Primary Scale of Intelligence (WPPSI-IV)[53,83], the "Picture memory subtest" and "Zoo subtest," were used to measure the working memories of children in the 1st- and 4th-rank children (n = 82). The picture-memory test required the child to memorize objects in a picture during a designated period and then to identify the items viewed within several objects after turning the page. In the Zoo test, the tester first placed animal cards at specific locations on a zoo map. After withdrawing the map, the child was then asked to return the animal cards to their correct positions, using only their

memory. The sum of the two scores was then converted into the working-memory index (WMI), which is the measure of each child's working-memory capabilities.

**Inventory data collection**. The teachers who worked with the children every day carried out the evaluations with respect to the children's daily behaviors in the classroom. A total of fifteen teachers carried out the evaluations (6 for the 82 children in behavioral studies and 9 for the 175 children in the inventory studies), and the chief teacher of each preschool helped to check for differences in ratings across the teachers and made the final decision.

*Social-dominance rating scale*. Dodge's teacher-rating scale was adopted for assessing the social-dominance levels of the young children[54], which was based on the Teacher Checklist of Dodge and Coie[84]. There were five items related to social dominance (Cronbach alphas = 0.89): "this child is a leader", "this child gets what he or she wants", "this child is competitive", "this child suggests to other children how things should be done", and "this child is frequently the center of the group". The teachers responded to each statement on a seven-point Likert scale, with "1" indicating "never" and "7" indicating "always". The five scores were averaged and represented a child's social-dominance score.

*Resource control strategy scale*. The teacher-rated Resource Control Strategy Scale[56] was used to assess the children's prosocial strategies of control (e.g., "This child promises friendship to get what s/he wants," "This child promises to do something in return to get what s/he wants"; $\alpha = 0.74$) and coercive strategies of control (e.g., "This child gets what s/he wants by bullying others," "This child gets what s/he wants by making verbal threats or threats of aggression"; $\alpha = 0.87$). Using a 7-point Likert scale, higher scores indicate higher endorsement of strategy employment. The teachers also rated the children's resource-control effectiveness (e.g., "This child usually gets first access to preferred toys when with peers," "This child usually plays with the favored toys when with peers"; $\alpha = 0.85$) (from hardly true to mostly true on the 7-point scale).

*Childhood executive functioning inventory*. The Working Memory subscale of the Chinese version of the Childhood Executive Functioning Inventory was adopted in this study[85]. The original version of this inventory was developed by Thorell and Nyberg[55] and then translated into different languages. The teachers were asked to evaluate the children's working-memory performance based on the children's daily behavior in the classroom. Ratings were made on a 5-point Likert scale, and a higher score indicated worse performance. The ratings were therefore scored in reverse for the following statistics.

**Event-related potential studies**. The ERP study was performed in a quiet room in the preschool. The children were told that they were selected to participate in computer games. Wearing a cap that conferred a special superpower (the EEG cap), their mission was to watch the photographs on the screen carefully. Children who completed the task could obtain mysterious gifts (animal-shaped biscuits). Three classes of facial expressions (dominant, subordinate, and neural) were built with peers of preschool children and based on studies that demonstrated dominant and submissive feelings evoked by specific facial features[58–60]. At the beginning of each trial, a fixation cue was presented for 500 ms to alert the children to the appearance of a facial photograph. Before the real test, 12 photographs (three trials for each type) were presented randomly as the practice session to familiarize the children with the entire procedure. Then, the photographs continuously showed up for 500 ms with a blank screen as the interstimulus interval for 800–1100 ms. Each subject needed to complete 90 trials (30 for each facial type) in a random order. In this study, all the children were finally told that they completed the task mission successfully and received gifts.

**Electrophysiological recording and processing**. Electroencephalogram (EEG) signals were recorded from 32 Ag-/AgCl–sintered electrodes with the SynAmp2 system (Quik-Cap Neo Net, Compumedics Ltd., VIC, Australia). The recording was referenced to the bilateral mastoids. A vertical electrooculogram (EOG) was recorded over the left eye, and a horizontal EOG was recorded over the left and right orbital rims. All impedance of the electrode was maintained below 5 kΩ throughout the recording. The acquisition-sample rate was 1000 Hz with a DC-100-Hz band-pass filter. The ERPs were analyzed with Curry 8 software (Compumedics Ltd., VIC, Australia). The brain signals were first filtered with a low-pass filter down 12 dB at 50 Hz and a high-pass filter down 12 dB at 1 Hz. Artifacts of EEG signals were corrected by using independent-component analysis (ICA)[86]. Then, all continuous EEG signals were epoched from 100 ms prestimulus (-100–0 as the baseline correction) to 900 ms poststimulus. Visual inspection was first performed to confirm the window range of FN400 (300–450 ms). The FN400 value was averaged within the time window per subject and per condition. The FN400 amplitudes of each facial expression were further corrected by subtracting the averaged amplitude of the neutral facial condition.

**Statistics and reproducibility**. All statistical analyses were completed using SPSS 22.0 or GraphPad Prism 8.0 software. Individual data are represented as the mean

± standard error of the mean (S.E.M.). The boxplots show the median values, with the box passing through the 25th–75th percentile. The top and bottom lines show the maximum and minimum lifespan values. For mice, the exploration time, discrimination index within cages, and winning rate for drug treatment were examined by Wilcoxon signed-rank tests. A two-sided unpaired $t$-test was applied to fEPSP slopes and the discrimination index for drug treatment. The correlation analyses were tested by Spearman correlation analysis. The relationships between social rank and drug treatment were tested by Fisher's exact test. The consistencies between social rank and gene expression were evaluated by the binary rank-correlation test based on the index $p$:

$$p = \frac{100 \times \sum i,j \begin{cases} 1, \Delta CT_i > \Delta CT_j \\ 0, \Delta CT_i \le \Delta CT_j \end{cases}}{n}$$

For each comparison, i had a higher social rank than j from the same cage. $\Delta CT$ represents the relative gene expression. The sample size n was calculated as 6 pairs times the cage number. The z score of each gene was approximated as $\frac{(p-0.5)}{\sqrt{p(1-p)/n}}$ and converted to the $p$-value.

For children, comparisons of memory scores or indices were examined by two-sided unpaired $t$-tests. Two-way ANOVA analyses were used in ERP studies. The correlation analyses were tested by Pearson correlation analysis. The mediation effect was examined by regression analysis followed by the Sobel test.

## Data availability

The datasets generated and/or analyzed for this study are deposited in Mendeley Data (https://data.mendeley.com/datasets/hgkb4hh5r5/draft?a=3c8498e1-83cf-4ed7-a5b7-187c697774b0).

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

## Acknowledgements

We thank the children who participated in this research and their parents for allowing them to do so, as well as the members of the Chou, Yang and Kuo labs for their help with the experiments. We thank Dr. Ming-Yueh Huang in the Institute of Statistical Sciences, Academia Sinica, for his great support in statistical analyses. We also thank Mr. Hsueh-Kai Chang at the Electrophysiology Core in the Institute of Biomedical Sciences, Academia Sinica, for his support. The work was supported by the Ministry of Science and Technology (MOST 107-2410-H-007-070-MY2 to Y-J. C.), (MOST 106-2320-B-001-013 and 107-2320-B-001-026-MY3 to S-B.Y.), and (MOST 108-2636-B-007-002 Young Scholar Fellowship to T-H. K.), as well as by the Higher Education Sprout Project funded by the Ministry of Education and Ministry of Science and Technology (Grants 107Q2721E1 and 108Q2721E1 of the Interdisciplinary Research Project to Y-J. C. and T-H. K.; and the Brain Research Center to T-H. K.).

## Author contributions

YJ Chou, SB Yang and TH Kuo designed the experiments; YJ Chou, YK Ma, YH Lu and WS Tasi performed the experiments; YJ Chou, YK Ma, YH Lu, JT King, WS Tasi, SB Yang and TH Kuo analyzed the data; and YJ Chou, JT King, SB Yang and TH Kuo wrote the paper.

## Competing interests

The authors declare no competing interests.
