## [Peer Review File · Communications Biology]

Reviewers' comments:

Reviewer #1 (Remarks to the Author):

General:

The manuscript performs analysis of working memory abilities in humans and mice of varying social ranks. While the analyses and the questions asked are very interesting, I come away uncertain whether the conclusions on conserved mechanisms in mice and humans can stand their ground. Many of the observations appear correlational and could be due to other factors leading to dominance. The mouse analysis similarly draws several conclusions that appear correlational and would benefit from a possible loss- or gain- of function experiment. For example, would a knockdown of one of the identified genes change dominance behavior? If the authors have the ability, they could consider that such a manipulation might strengthen their arguments. Furthering this, is there evidence for any of these genes showing variation in the human population? Overall the analyses come across as two separate stories, however the analyses and experiments appear to be sound. If the authors could clarify some of the risks and weaknesses of the proposed conclusions, it would strengthen the article.

Minor:

Is the reference to Figure S1 identical to the author's previously published analysis? In that case, it may be more suitable to simply reference the paper rather than re-publishing the analysis.

Figure 4G - I am not sure if drawing arrows between these different analysis outcomes is appropriate, we don't know which phenomenon leads to the next biologically.

Reviewer #2 (Remarks to the Author):

Chou and colleagues investigated the relationship between social status and memory ability in young mice and children. They found that mice and children of lower social rank showed poorer memory capability as indexed by performance on a novel object recognition task and a working memory task. Moreover, Chou et al. showed mice with stronger memory capability and of higher social rank showed augmented LTP and higher expression of various genes implicated in learning and plasticity such as NR2B.

The paper is nicely written and presents a clear and comprehensive story based on a comparative study. Such cross-species investigations are not common place in neuroscience. Thus I applaud the efforts of the authors. However, I have some concerns about the interpretation of the results and how development may potential confound the study's interpretations. I outline my concerns below.

- The authors study developing mice - 3-4weeks of age - as well as pre-school children (~4years of age). Although hippocampal memory has started to mature at this age (in both species), memory is by no means fully developed by even 4weeks of age in rodents (e.g. Campbell 1972). Specifically, infantile amnesia is thought to come to an end around 3weeks of age in rodents, but it takes several weeks for hippocampal memory to fully mature (particularly long-term memory). Thus, it may be that the effects Chou and colleagues observe in their study reflect the different levels of maturity of mice of low and high social rank - or perhaps the social ranking (in this developmental context) may reflect the degree of maturity of the mice. This interpretation is particularly strengthened by the observation that low ranking mice did not show reliably recognition of the novel object and it seems they did not seem to alternate above chance on the spontaneous alternation task. The LTP results could also be explained with this developmental perspective. If the authors want to convincingly show that social rank is tied to memory capability they need to test older mice.
- Moreover, the object recognition task is not the best task for testing hippocampal memory as the task is not strictly dependent on the hippocampus (e.g. Langston & Wood (2010)). This makes the interpretation of the results even more challenging.

Minor comments:

- For their LTP analysis, the authors say they wanted to assess if memory ability in mice of different social rank were caused by underlying differences in LTP. To be clear, the authors did a correlational not a causal analysis - they assessed if LTP differed in mice of different social rank.
- There are a number of typos in various places throughout the text.

Reviewers' comments:

Reviewer #1 (Remarks to the Author):

General:

1. - The manuscript performs analysis of working memory abilities in humans and mice of varying social ranks. While the analyses and the questions asked are very interesting, I come away uncertain whether the conclusions on conserved mechanisms in mice and humans can stand their ground. Many of the observations appear correlational and could be due to other factors leading to dominance. The mouse analysis similarly draws several conclusions that appear correlational and would benefit from a possible loss- or gain- of function experiment. For example, would a knockdown of one of the identified genes change dominance behavior? If the authors have the ability, they could consider that such a manipulation might strengthen their arguments.

Thanks for reviewer's suggestion. Instead of knockdown, we took a pharmacological approach to activate BDNF/TrkB/CREB signaling pathway using Sodium Butyrate and Rolipram. Both drugs are known to induce hippocampal LTP and to improve memory. Our new data suggested that administration of Sodium Butyrate or Rolipram intraperitoneally for 2 weeks increased both mice memory and social rank (Figure 4B-E), which further strengthen the link between these two phenotypes.

Figure 4. Memory-improving drug enhanced social dominance.

2. Furthering this, is there evidence for any of these genes showing variation in the human population? After literature searching, we did find some references identifying variants in *Gria1*, *Grin2B* and *Creb1*. We have added these information into the discussion (Line 278).

- Ludwig, K. U. et al. Variation in *GRIN2B* Contributes to Weak Performance in Verbal Short-Term Memory in Children With Dyslexia. *Am J Med Genet B* **153b**, 503-511 (2010).
- Jiang, Y. et al. Functional human *GRIN2B* promoter polymorphism and variation of mental processing speed in older adults. *Aging-U.S.* **9**, 1293-1306 (2017).
- Zubenko, G. S. et al. Sequence variations in *CREB1* cosegregate with depressive disorders in women. *Mol Psychiatr* **8**, 611-618 (2003).
- Kerner, B. et al. Polymorphisms in the *GRIA1* Gene Region in Psychotic Bipolar Disorder. *Am J Med Genet B* **150b**, 24-32, doi:10.1002/ajmg.b.30780 (2009).

3. Overall the analyses come across as two separate stories, however the analyses and experiments appear to be sound. If the authors could clarify some of the risks and weaknesses of the proposed conclusions, it would strengthen the article.

In this revised version, we clearly pointed out that, despite similarity in behavioral data, the neural mechanisms in human are still need to be studied (Line 275). We also emphasized that the causality between social hierarchy and memory ability remains to be explored (Line 319). Hope our discussion has clarified these two points clearly.

Minor:

4.- Is the reference to Figure S1 identical to the author's previously published analysis? In that case, it may be more suitable to simply reference the paper rather than re-publishing the analysis.

The social rank in Figure S1 is not from previous publication. It is a new data to demonstrate that the stable social rank is repeatable. We have changed the sentence from "Based on previous data" to "Similar to previous data" (Line 104).

5.- Figure 4G - I am not sure if drawing arrows between these different analysis outcomes is appropriate, we don't know which phenomenon leads to the next biologically.

The purpose of figure 6 (original Figure 4) is to explore the hypothesis that memory could help prosocial strategy learning to enhance social dominance. Although the regression analysis cannot directly reveal the causality among these factors, the result indicates the mediation effect of prosocial strategy, which is consistent with our hypothesis. We therefore have arrows in Figure 6G to illustrated our hypothesized model clearly.

In the result and figure legend, we have emphasized that figure 6G is our proposed model (Line 214). We have also pointed out that the casualty between memory and social dominance, in fact, remains to

be investigated (Line 319). However, if reviewer still thinks the presence of arrows are misleading, we could definitely remove them from the figure.

Reviewer #2 (Remarks to the Author):

Chou and colleagues investigated the relationship between social status and memory ability in young mice and children. They found that mice and children of lower social rank showed poorer memory capability as indexed by performance on a novel object recognition task and a working memory task. Moreover, Chou et al. showed mice with stronger memory capability and of higher social rank showed augmented LTP and higher expression of various genes implicated in learning and plasticity such as NR2B.

The paper is nicely written and presents a clear and comprehensive story based on a comparative study. Such cross-species investigations are not common place in neuroscience. Thus I applaud the efforts of the authors. However, I have some concerns about the interpretation of the results and how development may potential confound the study's interpretations. I outline my concerns below.

6.- The authors study developing mice - 3-4weeks of age - as well as pre-school children (~4years of age). Although hippocampal memory has started to mature at this age (in both species), memory is by no means fully developed by even 4weeks of age in rodents (e.g. Campbell 1972). Specifically, infantile amnesia is thought to come to an end around 3weeks of age in rodents, but it takes several weeks for hippocampal memory to fully mature (particularly long-term memory). Thus, it may be that the effects Chou and colleagues observe in their study reflect the different levels of maturity of mice of low and high social rank - or perhaps the social ranking (in this developmental context) may reflect the degree of maturity of the mice. This interpretation is particularly strengthened by the observation that low ranking mice did not show reliably recognition of the novel object and it seems they did not seem to alternate above chance on the spontaneous alternation task. The LTP results could also be explained with this developmental perspective. If the authors want to convincingly show that social rank is tied to memory capability they need to test older mice.

*Thanks for reviewer pointing this out. We completely ignored this important factor, therefore only presented the adult data in the supplemental data in the last version. In this revised version, we moved the adult behavioral data to main **Figure 2**. The adult LTP data is currently in main **Figure 3D-F**.*

Figure 2. Adult mice with higher rank showed better memory performance.

Figure 3. Higher rank mice showed greater LTP in hippocampal slices.

7.- Moreover, the object recognition task is not the best task for testing hippocampal memory as the task is not strictly dependent on the hippocampus (e.g. Langston & Wood (2010). This makes the interpretation of the results even more challenging.

Although NOR is not the best assay for hippocampal memory, the importance of hippocampus for NOR have been documented in several studies. We have added a few review papers in references (Line 268).

- Broadbent, N. J., Gaskin, S., Squire, L. R. & Clark, R. E. Object recognition memory and the rodent hippocampus. *Learn Memory* **17**, 794-800 (2010).
- Feinberg, L. M., Allen, T. A., Ly, D. & Fortin, N. J. Recognition memory for social and non-social odors: Differential effects of neurotoxic lesions to the hippocampus and perirhinal cortex. *Neurobiol Learn Mem* **97**, 7-16 (2012).
- Cohen, S. J. et al. The Rodent Hippocampus Is Essential for Nonspatial Object Memory. *Curr Biol* **23**, 1685-1690 (2013).
- Cohen, S. J. & Stackman, R. W. Assessing rodent hippocampal involvement in the novel object recognition task. A review. *Behav Brain Res* **285**, 105-117 (2015).
- Bird, C. M. The role of the hippocampus in recognition memory. *Cortex* **93**, 155-165 (2017).
- Cinalli, D. A., Cohen, S. J., Guthrie, K. & Stackman, R. W. Object Recognition Memory: Distinct Yet Complementary Roles of the Mouse CA1 and Perirhinal Cortex. *Front Mol Neurosci* **13** (2020).

However, we do agree that other brain regions, especially perirhinal cortex, might be more important for NOR memory. We have pointed this out in our discussion (Line 269). Thanks for reviewer's reminder.

Minor comments:

8.- For their LTP analysis, the authors say they wanted to assess if memory ability in mice of different social rank were caused by underlying differences in LTP. To be clear, the authors did a correlational not a causal analysis - they assessed if LTP differed in mice of different social rank.

Thanks for reviewer's reminder, we have corrected the sentence from "whether augmented LTP caused the enhanced memory ability in mice with a higher social rank" to "whether the enhanced memory ability in mice with a higher social rank is associated with augmented LTP"(Line 136).

9.- There are a number of typos in various places throughout the text.

The 1st version of manuscript was edited by professional editor (please find the Certificate below). We have further examined this new version multiple time. If reviewer think it is still necessary, we would be happy to send it out for the 2nd professional editing.

Editing Certificate

This document certifies that the manuscript

social hierarchy and memory

prepared by the authors

Tsung-Han Kuo

was edited for proper English language, grammar, punctuation, spelling, and overall style by one or more of the highly qualified native English speaking editors at AJE.

This certificate was issued on **March 1, 2021** and may be verified on the [AJE website](https://www.aje.com) using the verification code **9239-1F96-6861-D4DA-3AAP**.

Neither the research content nor the authors' intentions were altered in any way during the editing process. Documents receiving this certification should be English-ready for publication; however, the author has the ability to accept or reject our suggestions and changes. To verify the final AJE edited version, please visit our verification page at [aje.com/certificate](https://www.aje.com/certificate). If you have any questions or concerns about this edited document, please contact AJE at support@aje.com.

AJE provides a range of editing, translation, and manuscript services for researchers and publishers around the world.

For more information about our company, services, and partner discounts, please visit [aje.com](https://www.aje.com).

REVIEWERS' COMMENTS:

Reviewer #1 (Remarks to the Author):

The authors have addressed most of the comments satisfactorily.

Reviewer #2 (Remarks to the Author):

I am satisfied with the edits made by the authors to the revised manuscript. I only have minor comments to add which mostly relate to textual/grammatical errors and typos. I list these below. I would strongly recommend the authors to carefully proof read the manuscript before publication.

Otherwise, I would be happy to endorse publication.

- fig1 G,H: y-axis label should say alternation not alternative, same goes for y-axis in fig2
- the authors need to be cautious when reporting results that are not statistically significant. I would not say a result is close to significance if it is not statistically significant, rather say 'we observed a trend for'

- line 22: can should be could

- line 95: 'Improving mice memory by Sodium Butyrate (SB) or Rolipram can' This sentence does not read right. Rather say 'improving mouse memory by Sodiam Butyrate (SB) or Rolipram was also found to....' also enhanced dominant status.

- line 11: significant should be significance

- line 278 challenge should be challenging

- line 276, human should be humans

- line 325: ' Similarly, stressful environments, such as less social interaction, less attention from parents/teachers/peers or fewer resources...' This sentence does not read well, I suggest changing it to 'Similarly, stressful environments, such as those associated with reduced social interaction, or attention from parents/teachers/peers or limited availability of resources....'

REVIEWERS' COMMENTS:

Reviewer #1 (Remarks to the Author):

The authors have addressed most of the comments satisfactorily.

Thanks for reviewer's comments to push us to improve the manuscript.

Reviewer #2 (Remarks to the Author):

I am satisfied with the edits made by the authors to the revised manuscript. I only have minor comments to add which mostly relate to textual/grammatical errors and typos. I list these below. I would strongly recommend the authors to carefully proof read the manuscript before publication.

Otherwise, I would be happy to endorse publication.

- fig1 G,H: y-axis label should say alternation not alternative, same goes for y-axis in fig2
- the authors need to be cautious when reporting results that are not statistically significant. I would not say a result is close to significance if it is not statistically significant, rather say 'we observed a trend for'
- line 22: can should be could
- line 95: 'Improving mice memory by Sodium Butyrate (SB) or Rolipram can' This sentence does not read right. Rather say 'improving mouse memory by Sodiam Butyrate (SB) or Rolipram was also found to....'
also enhanced dominant status.
- line 11: significant should be significance
- line 278 challenge should be challenging
- line 276, human should be humans
- line 325: ' Similarly, stressful environments, such as less social interaction, less attention from parents/teachers/peers or fewer resources...' This sentence does not read well, I suggest changing it to 'Similarly, stressful environments, such as those associated with reduced social interaction, or attention from parents/teachers/peers or limited availability of resources....'

Thanks for reviewer's careful examinations. We have corrected the above mistakes. In addition, we have the manuscript edited again by the profession editing (see attached) and further checked by all three corresponding authors.

This document certifies that the manuscript
social hierarchy and memory

prepared by the authors
Tsung-Han Kuo

was edited for proper English language, grammar, punctuation, spelling, and overall style by one or more of the highly qualified native English speaking editors at AJE.

This certificate was issued on **December 29, 2021** and may be verified on the AJE website using the verification code **9239-1F96-6861-D4DA-3AAP**.

Neither the research content nor the authors' intentions were altered in any way during the editing process. Documents receiving this certification should be English-ready for publication; however, the author has the ability to accept or reject our suggestions and changes. To verify the final AJE edited version, please visit our verification page at aje.com/certificate. If you have any questions or concerns about this edited document, please contact AJE at support@aje.com.